# Establishment of a QuEChERS-UPLC-MS/MS Method for Simultaneously Detecting Tolfenpyrad and Its Metabolites in Tea

**Zihan Wang** [1,†], **Xinru Wang** [1,2,†], **Min Wang** [1,3], **Ziqiang Li** [1,4], **Xinzhong Zhang** [1,2], **Li Zhou** [1,2], **Hezhi Sun** [1,2], **Mei Yang** [1,2], **Zhengyun Lou** [1,2], **Zongmao Chen** [1,2] and **Fengjian Luo** [1,2,*]

1. Tea Research Institute, Chinese Academy of Agricultural Sciences, Hangzhou 310008, China
2. Key Laboratory of Biology, Genetics and Breeding of Special Economic Animals and Plants, Ministry of Agriculture and Rural Affairs, Hangzhou 310008, China
3. College of Horticulture and Landscape, Tianjin Agricultural University, Tianjin 300384, China
4. College of Plant Protection, Jilin Agricultural University, Jilin 130118, China
* Correspondence: lfj@tricaas.com; Tel.: +86-571-86650624; Fax: +86-571-86650331
† These authors contributed equally to this work.

**Abstract:** An analytical method simultaneously detecting pesticide and its metabolites, especially with higher toxicity, was urgently needed for supervision and safety evaluation of agricultural products. In the present study, a modified QuEChERS method coupled with a ultra-high performance liquid chromatography-tandem mass spectrometry (UPLC-MS/MS) for simultaneous determination of tolfenpyrad (TFP) and its metabolites in tea matrices (fresh tea shoots, green tea, black tea, green tea infusion and black tea infusion) was established. The method validation showed good linearity (correlation coefficients $\geq$ 0.9945), accuracy (recoveries in 75.38–109.90%), sensitivity (limits of quantification $\leq$ 0.05 mg kg$^{-1}$), and precision (relative standard deviations $\leq$ 19.09%). The established method was then applied to detect 40 market samples, resulting in 60.0% positive rate of TFP, besides, two metabolites including PT-CA, which is more toxic than the parent TFP, and PT-OH were also detected in the samples with high TFP residue ($\geq$0.048 mg kg$^{-1}$). The method established in the present work was thus of significant importance in comprehensive monitoring and of TFP in tea products.

**Keywords:** pesticide residue; analytical method; metabolites; tea

## 1. Introduction

Tea is one of the most popular beverages globally, attributed to its charming aroma and health-promoting benefits [1]. In addition, as an important cash crop in China, the total tea output of tea has risen from 2.31 to 3.18 million tons with an average annual increasement of 7.5% over the past five years [2]. The great majority of tea plants (*Camellia sinensis* L.) grow in warm and humid environment, which may result in 7–10% loss in yield under the attack of pests and diseases [3]. By now, chemical control is still the most effective approach to protect tea plants from pests and pathogens.

TFP, (Table 1), a pyrazole insecticide, was firstly registered in Japan in 2003 [4]. Its mode of action is inhibiting the electron transfer of mitochondrial complex I to prevent the oxidative phosphorylation process in energy metabolism. TFP has been shown to be effective against wide range of insect pests such as those in the *Lepidoptera, Hemiptera, Coleoptera, Diptera, Thysannoptera*, and *Acarina* families [5]. Shivaleela and Chowdary's study showed that after spraying TFP as 150 g a.i. ha$^{-1}$ in cucumber, the mortality of thrips, red pumpkin beetles and leafhoppers were up to 91.16%, 70.99% and 92.41%, respectively, with a control efficacy significantly superior over imidacloprid, fipronil, and chlorpyriphos [6]. Mallick et al. reported a 88.1–100.0% reduction of pests including jassid, thrips and aphid





after TFP application in an okra farm [7]. Similarly, Buzza reported an excellent control effect of TFP against Colorado potato beetle [8]. Because of its high-efficiency, broad-spectrum and low infusion factor during tea brewing, which is based on its low water solubility (0.087 mg L$^{-1}$), TFP has been registered for the control of tea lesser leafhopper and widely used in tea plants in China in recent years.

Primarily, metabolic reactions of pesticides in plants usually including three phases, Phase I metabolism, Phase II conjugation, and Phase III transportation [9]. As far as we know, metabolism research on TFP is limited so far. According to the Joint Meeting on Pesticide Residues (JMPR) evaluation report [4,10], TFP was relatively easy to be absorbed by cabbage, producing the major metabolites including OH-PT, OH-T-CA, OH-T-OH and CA-T-AM. As for peach plants, the main metabolites in leaves were PT-CA, CA-T-CA, T-CA and the corresponding conjugates. Besides, Koji found that the concentration of PT-CA was up to 2.88 ± 0.04 mg L$^{-1}$ in the plasma from a corpse died of a suspected TFP poisoning case [11]. What's more, as shown in Table 1, the acute oral toxicity of PT-CA to rats is stronger than that of its parent TFP. Thus, in view of the widespread usage of TFP on tea plants, a comprehensive analytical method for simultaneously detection of TFP and its metabolites in tea is necessary.

Nowadays, the main sample pretreatment techniques of pesticide residue analysis include the liquid-liquid extraction (LLE), solid phase extraction (SPE), solid-phase micro extraction (SPME), gel permeation chromatography (GPC) and QuEChERS (Quick, Easy, Cheap, Effective, Rugged and Safe) [12]. Our preliminary experiment found that, instead of the SPE method by using TPT cartridge, QuEChERS method represented better efficiency for simultaneously determination of both TFP and its metabolites. The QuEChERS method was first published by Anastassiades and Lehotay in 2003 [13]. Originally, this approach was developed for sample-preparation of pesticides analysis in fruits and vegetables. Then owing to the inherent advantages and green chemistry characteristics, the QuEChERS quickly expanded its application to various compounds, including pharmaceuticals, myco-toxins, and polycyclic aromatic hydrocarbons in a wide variety of complex matrices [14]. Moreover, great flexibility of QuEChERS has provided more possibilities for its improvements and modifications. Now there are three main compared versions, one based on the original unbuffered method, and the other two used buffering agent to ensure more efficient extractions of pH-dependent analytes as well as less degradation of labile analytes [15]. However, the strong matrix interference may cause a lack of selectivity towards target compounds, so it's an imperative of further cleanup steps by utilizing sorbents such as C$_{18}$, primary secondary amine (PSA) and graphitized carbon black (GCB) [16,17].

In this study, a modified QuEChERS method combined with UPLC-MS/MS to simultaneously detect TFP and its four main metabolites (PT-OH, PT-CA, OH-T-CA and CA-T-CA) in five tea matrices, including fresh tea shoots, green tea, black tea, green tea and black tea infusion was established. Then the developed method was applied to accurately analyze the residue of TFP as well as its metabolites in market samples. The method is important for supervision as well as comprehensive risk assessment of TFP with its metabolites taken into consideration.

**Table 1.** The chemical structure and toxicological parameters of TFP and its main metabolites.

| Analytes | Structure [4,10] | Molecular Formula | Molecular Weights (Da) | LD$_{50}$ [10] (mg kg$^{-1}$ Rats) |
|---|---|---|---|---|
| TFP |  | $C_{21}H_{22}ClN_3O_2$ | 383.87 | 386 |

**Table 1.** *Cont.*

| Analytes | Structure [4,10] | Molecular Formula | Molecular Weights (Da) | LD$_{50}$ [10] (mg kg$^{-1}$ Rats) |
|---|---|---|---|---|
| PT-OH |  | $C_{21}H_{22}ClN_3O_3$ | 399.14 | - |
| PT-CA |  | $C_{21}H_{20}ClN_3O_4$ | 413.11 | 27.4 |
| OH-T-CA |  | $C_{14}H_{12}O_4$ | 244.07 | 2024 |
| CA-T-CA |  | $C_{14}H_{10}O_5$ | 258.07 | >2000 |

## 2. Materials and Methods

### 2.1. Materials and Reagents

Standards of TFP and CA-T-CA were from Anqu Chemistry Co., Ltd. (Shanghai, China). PT-OH, PT-CA and OH-T-CA standards were entrusted to company for synthesis. HPLC grade Acetonitrile (MeCN) and methanol (MeOH) were purchased from Merck (Darmstadt, Germany). Formic acid (FA, HPLC, ≥99.0%) and analytical reagent (AR) ammonium hydroxide solution (AHA, 25–28%) were from Macklin Biochemical Co., Ltd. (Shanghai, China). Ammonium acetate (AA, HPLC, ≥98.0%) was from ANPEL Laboratory Technologies (Shanghai, China). Acetic acid (HAC, AR, ≥99.5%) and magnesium sulfate anhydrous (MgSO$_4$, AR, ≥98.0%) were obtained from Lingfeng Chemistry Reagent Co., Ltd. (Shanghai, China). Cleaner polymer weak anion exchange (PWAX, 40–60 μm), strong cation exchange (SCX, 40–60 μm), C$_{18}$-N (40–60 μm), C$_{18}$, (40–60 μm), primary-secondary amine (PSA, 40–63 μm) and graphitized carbon black (GCB, 120–400 mesh) were purchased from Bonna Agela Technologies Co., Ltd. (Tianjin, China). Hydroxyl carbon nanotube (C$_{NT}$-OH, 10–20 nm) was purchased from Timesnano Organic Chemicals Co., Ltd. (Chengdu, China).

### 2.2. Preparation of Compound Standard Solutions

Following the preparation of separate stock standard solutions of each component (1000 mg L$^{-1}$) by weighing and dissolving in MeOH, an appropriate volume of each individual stock standard was combined to create a mixed standard solution (100 mg L$^{-1}$). Prior to use, all of the standard solutions were kept at −20 °C and shielded from light. In order to provide matrix-matched standards and spike samples for the validation experiments, the mixed working standard solutions were freshly configured within the concentration range of 0.01–5 mg L$^{-1}$ by dilution with MeCN.

### 2.3. Sample Preparation

The blank fresh tea shoots samples used for method establishment were obtained from a tea plantation at Hangzhou (30.3° N, 120.2° E, Zhejiang, China). Green tea and black tea samples were made from the collected fresh tea shoots referring to the following standard steps: as for green tea, spreading (indoor temperature, 2–3 h), deenzyming (220 °C, 5 min), rolling (40 r min$^{-1}$, 30 min), drying (100 °C, 60 min); for black tea, withering (indoor temperature, 12 h), rolling (40 r min$^{-1}$, 30 min), fermenting (38 °C, 8 h), drying (100 °C,

60 min). Tea infusion was prepared according to the methodology for sensory evaluation of tea in China (Chinese National Standard GB/T 23776-2018). In brief, ground dry tea was brewed with boiling water in the ratio of 1/50 (*v/v*) for 5 min, and the first infusion (F1) was obtained by filtering the water extract, then the brewing step was repeated twice to gain the second (F2) and the third infusion (F3).

For fresh tea shoots and dry tea (green and black tea) samples, 2.0 g ground fresh tea shoots or 1.0 g ground dry tea sample was weighted into 50 mL centrifuged tube and soaked with 5 mL water for 30 min. Then 10 mL of 1% formic acid in MeCN (1%-FA-MeCN) was added, the mixture was shaken vigorously for 5 min and incubated for 2 h. Next, 5.0 g NaCl (3.0 g for dry tea) was added into the extract solution, after vortexed for 3 min and centrifuged for 5 min at 10,000 r $min^{-1}$, all the supernatant was transferred into a 50 mL centrifuged tube. The extraction steps were repeated one more time. After that, 7.5 mL of the combined extracting solution was evaporated to dryness and reconstituted in 1.5 mL MeCN. At last, all the MeCN extract was transferred into a 2 mL micro-centrifuge tube, containing 20 mg GCB, 20 mg $C_{NT}$-OH, 50 mg $C_{18}$ and 50 mg $MgSO_4$, vortexed homogeneously and centrifuged for 5 min at 12,000 r $min^{-1}$, then the supernatant was filtered (0.22 μm) before UPLC-MS/MS analysis.

For tea infusion (green and black tea infusion) samples, the 20 mL infusion sample was extracted with 20 mL 1%-FA-MeCN. Following vortexed vigorously, 8 g NaCl was added, and the mixture was further vortexed. Subsequently, the extraction was centrifuged for 5 min at 10,000 r $min^{-1}$, then all of supernatant was transferred into a 50 mL centrifuged tube. The extraction was repeated once more. At last, the 20 mL combined extract was evaporated to dryness, redissolved in 1 mL MeCN and filtered through a 0.22 μm membrane filter for UPLC-MS/MS analysis.

*2.4. UPLC-MS/MS Conditions*

The determination of all the samples was performed on a Waters Acquity UPLC in tandem with a Waters Xevo TQ-S Micro triple-quadrupole mass spectrometer (Waters, Milford, MA, USA) equipped with Electrospray Ionization (ESI) probe. Separation was achieved on an ACQUITY UPLC HSS T3 column (100 mm × 2.1 mm, 1.8 μm; Waters, Milford, MA, USA) at 40 °C. The 0.1% FA-MeOH and 0.1% FA-$H_2O$ were developed as mobile phases A and B respectively. The gradient program was described by the change of A as follow: 10–65% A at 0–1.5 min, 65–85% A for 3.5 min, 85–99% A for 2.5 min, 99–100% A for 1.5 min, maintained until the 9.8th min, 100–10% A for 0.5 min, and the total run time was 13 min, at a flow rate of 0.2 mL/min, with the injection volume of 5 μL.

Multiple reaction monitoring (MRM) mode was applied for the determination of target compounds, with a capillary voltage of 3.5 kV, source temperature of 150 °C, and desolvation temperature of 350 °C. The cone gas ($N_2$, 99.5%) and desolvation gas ($N_2$, 99.5%) were set at 50 L $r^{-1}$, 650 L $r^{-1}$, respectively. The optimum MRM parameters for TFP and its metabolites were shown in Table 2. During the usage of the instruments, manufacturers' user manual and instructions were strictly followed.

**Table 2.** The optimum MRM parameters for TFP and its metabolites.

| Compounds | Mode | Retention Time (min) | Precursor (*m/z*) | Cone Voltage (V) | Daughter Ions (Collision Energy, V) | |
|---|---|---|---|---|---|---|
| | | | | | Quantitation | Confirmation |
| TFP | ESI$^+$ | 9.00 | 384.20 | 35 | 197.20 (40) | 170.90 (25) |
| PT-OH | ESI$^+$ | 6.95 | 400.11 | 30 | 382.14 (10) | 213.04 (35) |
| PT-CA | ESI$^+$ | 7.37 | 414.12 | 25 | 227.05 (20) | 117.04 (40) |
| OH-T-CA | ESI$^-$ | 5.45 | 243.07 | 35 | 123.09 (25) | 199.27 (10) |
| CA-T-CA | ESI$^-$ | 5.88 | 256.89 | 25 | 137.03 (25) | 212.96 (10) |

### 2.5. Method Validation

The blank matrix of fresh tea shoots, dry tea, tea infusion were obtained according to the sample pretreatment method in Section 2.3. The linearity was estimated by matrix-matched calibration with gradient concentration matrix standard solutions (0.01, 0.05, 0.1, 0.5, 1 and 5 mg L$^{-1}$). The accuracy and precision of the method were estimated with recovery experiments for each matrix in five replicates at four spiked levels (0.005, 0.05, 1 and 10 mg L$^{-1}$ for fresh tea shoots and dry tea, 0.0005, 0.005, 0.01, 0.1 mg L$^{-1}$ for tea infusion). Referring to the intensity of the minimum concentration level in linear range, the signal-to-noise *(S/N)* of 3 was used for definition of the limit of detection (LODs) [18]. The limit of quantitation (LOQs) were defined as the minimum spiked level to meet the requirements of recovery and relative standard deviations (RSDs). The matrix effect (ME) was calculated by formula [19,20]: ME (%) = $(a_{matrix} - a_{solvent})/a_{solvent}$, where $a_{matrix}$ and $a_{solvent}$ were the slope of the standard curves for matrix-matched calibration and solvent-only calibration, respectively.

### 3. Results and Discussion

#### 3.1. Optimization of UPLC-MS/MS Conditions

#### 3.1.1. Optimization of MS/MS Conditions

In residue analysis of TFP and its metabolites, tandem mass detector with high selectivity and sensitivity provides an effective solution. For the most pesticides, the mass of the precursor ion corresponds to the mass of the compound's protonated molecule [M + H]$^+$ or [M + H]$^-$. Generally, there is an optimum cone voltage to cause a maximum sensitivity [21]. In method development, single standard solution, at 1 mg L$^{-1}$ in MeOH:H$_2$O (1:1, *v/v*), were infused directly into the ESI source at 10 μL min$^{-1}$ in the full scan mode. As a result, TFP, PT-OH and PT-CA got higher [M + H]$^+$ (*m/z* 384.20, 400.11, 414.12) intensity in positive electrospray ionization mode (ESI$^+$), while OH-T-CA and CA-T-CA exhibited higher intensity in negative electrospray ionization mode (ESI$^-$) (*m/z* 243.07, 256.88). Then a series of voltages (5, 10, 15, 20, 25, 30, 35, 40, 45, 50 V) were used for cone voltage optimization, as shown in Suppl. Figure S1, TFP, PT-OH, PT-CA, OH-T-CA and CA-T-CA obtained the highest intensity under the cone voltage of 35, 30, 25, 35, 25 V, respectively.

Under the optimum cone voltage, the optimum collision energy for MRM transitions for each compound was performed. For each parent ion, two fragment transitions were selected, one for quantitation and another for confirmation. By exerting various collision energy (5, 10, 15, 20, 25, 30, 35, 40, 45, 50 V) under MS/MS mode, the results were shown in Suppl. Figure S2, which illustrated that TFP mainly produced fragment ions of *m/z* 197.20 and *m/z* 170.90 under the optimal collision energy of 40 V and 25 V, respectively; the primary fragment ions *m/z* 382.14 and *m/z* 213.04 of PT-OH could obtain the highest intensity under collision energy at 10 V and 35 V; PT-CA mainly produced fragment ions of *m/z* 227.05 and *m/z* 117.04 with the corresponding optimal collision energy of 20 V and 40 V, respectively; as for OH-T-CA, the primary fragment ions *m/z* 199.27, *m/z* 123.09 exhibited the highest intensity under 10 V, 25 V, respectively; CA-T-CA produced main fragment ions *m/z* 212.96, *m/z* 137.03 with the optimal collision energy of 10 V, 25 V, respectively. In summary, the optimized MRM parameters for TFP and its metabolites were acquired and shown in Table 2.

#### 3.1.2. Optimization of Chromatographic Conditions

For LC-MS, the composition of the mobile-phases has a significant impact in ionization efficiency [22]. Owing to the good quality of chromatographic separation and analyte ionization, MeOH and MeCN are used as organic mobile phases [23,24]. In addition, the pH of mobile phase also plays an important role in the chromatographic retention for compounds with acid-base properties [25]. In order to improve the chromatographic separation and get higher MS intensity, different mobile phase combinations (A+B) were tested: 0.1% FA-MeOH + 0.1% FA-H$_2$O; 0.1% FA-MeOH + 10 mmol L$^{-1}$ AA-H$_2$O; 0.1% FA-MeCN + 10 mmol L$^{-1}$ AA-H$_2$O and 0.1% FA-MeCN + 0.1% FA-H$_2$O. As shown

in Suppl. Figure S3, TFP (384.2 > 197.2), PT-OH (382.14 > 197.07), PT-CA (414.12 > 227.05), OH-T-CA (243.07 > 199.27), CA-T-CA (256.89 > 137.03) all presented the optimal intensity with 0.1% FA-MeOH and 0.1% FA-H$_2$O been used. Hence, 0.1% FA-MeOH and 0.1% FA-H$_2$O were selected as the mobile phases (A+B) in the present study.

### 3.2. Optimization of Samples Pretreatment

3.2.1. Selection of Extractants

Given that some metabolites with relatively high water solubility, the extraction effect may be improved if the sample could be soaked in water in advance [26]. Therefore, different soaking solvents (H$_2$O, 1% FA-H$_2$O, 5% FA-H$_2$O, 1% HAC-H$_2$O and 5% HAC-H$_2$O) were tested. In Suppl. Figure S4, the extraction recoveries of the five compounds in both water and acidified water were greater than 80%, with the relative standard deviations (RSDs) less than 20%. To establish a method in simple economic terms, water was finalized as the socking solvent.

MeCN, in which salt alone can be applied to get a gratifying separation from water, is regarded as a common organic solvent for the QuEChERS [13]. Therefore, the extraction efficiencies for TFP and its 4 metabolites in MeCN, acidified MeCN (1% FA-MeCN, 2% FA-MeCN, 1% HAC-MeCN, 2% HAC-MeCN) and alkalized MeCN (2% AHA-MeCN) were compared. As presented in Figure 1, as MeCN was used, the recoveries of most compounds were better than 78.66%, except CA-T-CA, which was only 52.95%. Obviously, the intensity and recoveries of the target compounds reached the best when 1% FA-MeCN was applied. It is speculated that the result was induced by the different pHs among the five analytes. According to previous measurement, standard solutions of TFP and PT-OH exhibited mildly acidic, and the other metabolites showed stronger acid, in particular CA-T-CA, which has two carboxylic acid groups. Studies has also illustrated that acidic compounds could be extracted more effectively in acidified MeCN [27,28]. In summary, 1% FA-MeCN was finalized as the extraction solvent in the present study.

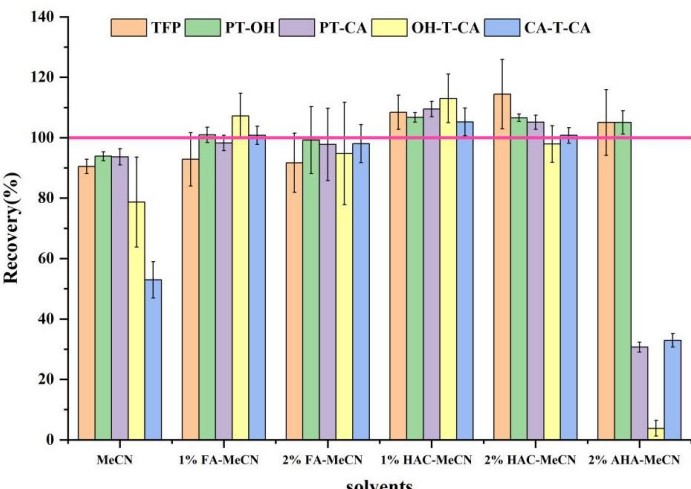

**Figure 1.** The recoveries of TFP and its metabolites in different extraction solvents (the red horizen line represents the recovery of 100%, and the closer to it, the higher the accuracy).

3.2.2. Optimization of Purification Conditions

As the key steps of QuEChERS, extraction and purification must balance the recovery rates of most target compounds and the purification effects [29]. In this work, the adsorbility of six common adsorbents (PWAX, SCX, C$_{18}$, PSA, C$_{18}$-N, and MgSO$_4$) with a series of dosages (10 mg, 20 mg, 50 mg, 100 mg and 200 mg) for TFP and its metabolites in 1 mg L$^{-1}$ standard solutions were tested. As shown in Suppl. Figure S5, C$_{18}$ (50 mg) and MgSO$_4$ (50 mg) were chosen without significant interferences to all analytes. However, as we know, tea represents a complex matrix, containing a large number of caffeine,

pigments, polyphenols, and other components [30]. As presented in the vial 1 of the Suppl. Figure S6, the crude MeCN extracts of green tea contained so much matrix co-extracts. So that the cleanup performance of $C_{18}$ and $MgSO_4$ were not good enough (vial 2 in the Suppl. Figure S6). In that case, GCB and $C_{NT}$-OH, two adsorbents with strong adsorption capacity, were tested for cleanup effects. GCB represents a carbon material with a strong affinity towards planar molecules, which can effectively removes pigments and sterols [31]. $C_{NT}$-OH, a new adsorbent, shows a strong adsorption capacity for pollutants like phenol and its derivatives [32]. In the following, different dose combinations of GCB and $C_{NT}$-OH were tested. In Figure 2, as 20 mg GCB and 20 mg $C_{NT}$-OH were applied, the recoveries for all the compounds could ranged in 77.33–88.34%, in the range of 77.33–88.34%. The purifying effect also met the instrument detection requirements (vial 3 in Suppl. Figure S6). The typical chromatogram was shown in Figure 3. In summary, 50 mg $C_{18}$ + 50 mgMgSO$_4$ + 20 mg GCB + 20 mg $C_{NT}$-OH were applied to clean up the extract of fresh tea shoots and dry tea matrices in present study.

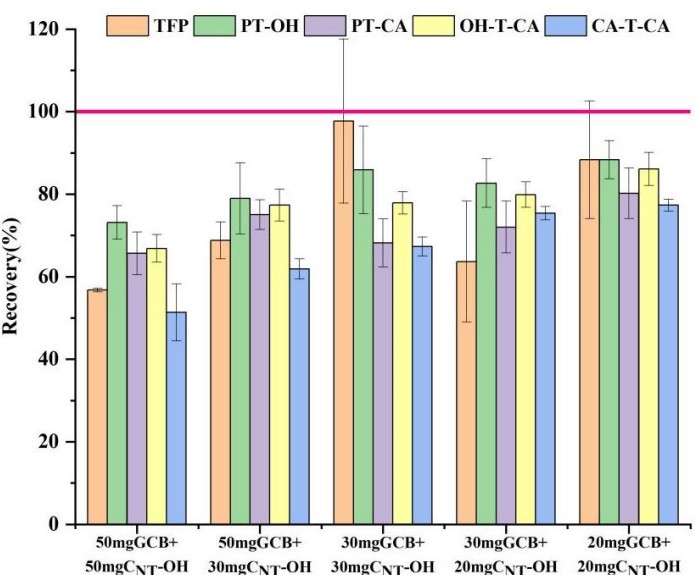

**Figure 2.** The recoveries of TFP and its metabolites under different dosage combinations of GCB and $C_{NT}$-OH (the red horizen line represents the recovery of 100%, and the closer to it, the higher the accuracy).

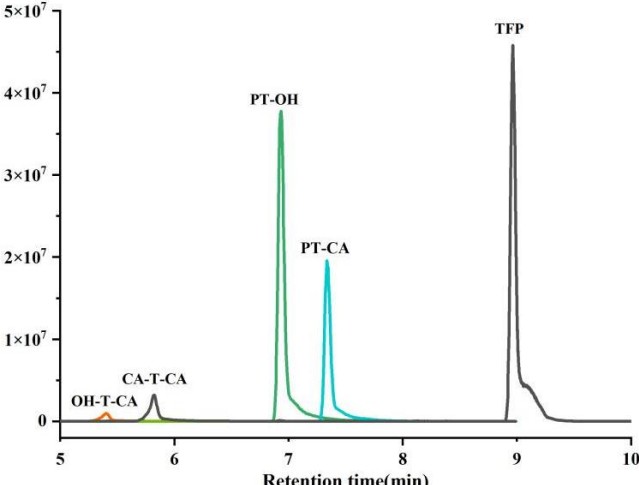

**Figure 3.** An overlay of MRM chromatograms of TFP and its metabolites in the green tea matrix-matched standard (2.5 mg kg$^{-1}$) acquired using the optimised MS/MS method.

As for tea infusion, clear extract (vial 4 in Suppl. Figure S6) could be detected directly. By instrumental analysis, the recoveries for all target compounds ranged in 78.36–92.32%, with RSDs $\leq$ 20.18%. Thus, the pretreatment method for tea infusion was finalized as described in Section 2.3.

### 3.3. Method Validation

#### 3.3.1. Linear, Matrix Effects and Limit of Detection

Under the optimized conditions, the methodology was verified. Linearity was evaluated by a series of matrix-matched calibration standards of TFP, PT-OH, PT-CA, OH-T-CA and CA-T-CA, respectively. As shown in Table 3, good linearities were found for TFP, PT-OH and PT-CA in the range of 0.01–5 mg L$^{-1}$ with the correlation coefficient over 0.9945, and 0.05–5 mg L$^{-1}$ for OH-T-CA and CA-T-CA with the correlation coefficient over 0.9987. The LODs of TFP and its metabolites in fresh tea leaves, green tea, black tea, green tea infusion and black tea infusion matrices were less than 0.01 mg L$^{-1}$.

**Table 3.** The linear equations, correlation coefficient ($R^2$), matrix effects (MEs) and limits of detection (LODs) of TFP and its metabolites in different tea matrices.

| Compound | Matrix | Linear Range (mg L$^{-1}$) | Regression Equation | $R^2$ | ME$_S$ | LOD$_S$ (mg kg$^{-1}$, mg L$^{-1}$) |
|---|---|---|---|---|---|---|
| TFP | Solvent | 0.01–5 | y = 375128x + 50660 | 0.9929 | - | 0.001 |
| | Fresh tea shoots | | y = 371479x + 30609 | 0.9991 | −0.010 | |
| | Green tea | | y = 219892x + 30895 | 0.9983 | −0.414 | |
| | Black tea | | y = 280541x − 4411.3 | 0.9996 | −0.252 | |
| | Green tea infusion | | y = 300626x + 29304 | 0.9972 | −0.199 | |
| | Black tea infusion | | y = 477403x + 73190 | 0.9974 | 0.273 | |
| PT-OH | Solvent | 0.01–5 | y = 174050x + 27174 | 0.9900 | - | 0.001 |
| | Fresh tea shoots | | y = 205091x + 18637 | 0.9975 | 0.178 | |
| | Green tea | | y = 160190x + 1590 | 0.9965 | −0.080 | |
| | Black tea | | y = 153899x − 1186.3 | 1.0000 | −0.116 | |
| | Green tea infusion | | y = 128895x + 28392 | 0.9973 | −0.259 | |
| | Black tea infusion | | y = 187879x + 27253 | 0.9972 | 0.079 | |
| PT-CA | Solvent | 0.01–5 | y = 96510x + 8175.9 | 0.9945 | - | 0.001 |
| | Fresh tea shoots | | y = 139557x + 11137 | 0.9967 | 0.446 | |
| | Green tea | | y = 83340x + 10047 | 0.9968 | −0.360 | |
| | Black tea | | y = 88994x − 3787.9 | 0.9994 | −0.078 | |
| | Green tea infusion | | y = 68545x + 13399 | 0.9976 | −0.290 | |
| | Black tea infusion | | y = 127623x + 14235 | 0.9983 | 0.322 | |
| OH-T-CA | Solvent | 0.05–5 | y = 5301.5x + 236.85 | 0.9994 | - | 0.01 |
| | Fresh tea shoots | | y = 5063.5x + 72.438 | 0.9994 | −0.045 | |
| | Green tea | | y = 2144.6x − 48.914 | 0.9991 | −0.595 | |
| | Black tea | | y = 2216.5x − 178.53 | 0.9987 | −0.582 | |
| | Green tea infusion | | y = 2972.4x + 108.24 | 0.9993 | −0.439 | |
| | Black tea infusion | | y = 4213.9x + 217.77 | 0.9997 | −0.205 | |
| CA-T-CA | Solvent | 0.05–5 | y = 35422x − 3072.5 | 0.9998 | - | 0.01 |
| | Fresh tea shoots | | y = 40081x + 1318.8 | 0.9990 | 0.131 | |
| | Green tea | | y = 17754x + 575.06 | 0.9998 | −0.499 | |
| | Black tea | | y = 18298x − 598.62 | 0.9996 | −0.483 | |
| | Green tea infusion | | y = 25330x + 3256.6 | 0.9998 | −0.285 | |
| | Black tea infusion | | y = 32730x + 1845.6 | 0.9994 | −0.076 | |

The matrix effect (ME) is the total influence of all sample components rather than analytes on measurement, which can inhibit or enhance the analyte signal due to co-eluted matrix components [33]. ME was categorized as low (−20%~20%), moderate (−50%~50%), strong (<−50% or >50%) [34,35]. In this study, the MEs were verified by comparing matrix-matched standards to solvent-based standards. As shown in Figure 4, the values of ME for 5 components in 6 matrices ranged from −50% to 50% except OH-T-CA, which represented MEs of −59.5% and −58.2% in the green and black tea matrices, respectively. It is hypothesized that some particular components in tea substrates could suppress the OH-T-CA signal intensity in ESI$^-$ mode, in spite of a purification step. This was in line

with other researchers' findings, which reported that the majority of pesticides had matrix suppression effects [36,37]. As a result, a series of matrix-matched standards were utilized for quantification in the current study.

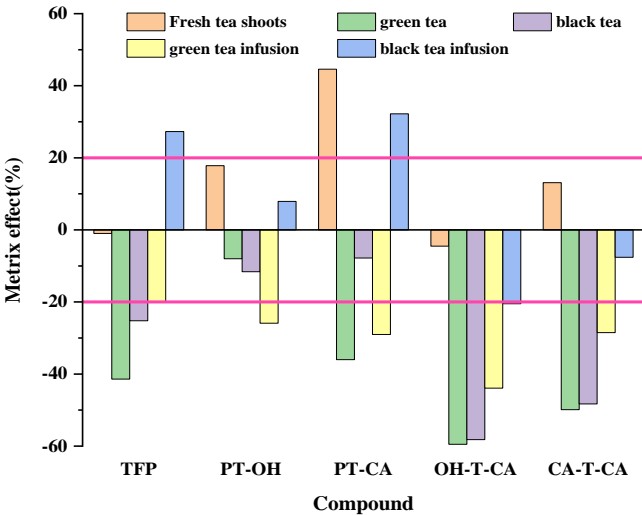

**Figure 4.** The matrix effect (%) of TFP and its metabolites in different tea matrix (the red horizen line represent the low matrix effect, ranging in −20%~20%).

### 3.3.2. The Accuracy and Precision

To evaluate the accuracy and precision of the established method for the residue analysis of TFP and its metabolites, the recovery test at four spiked levels (*n* = 5) were carried out. As shown in Table 4, the recoveries and the relative standard deviations (RSDs) were satisfactory for the pesticide residue analysis, accounting for 75.38–109.90% and 1.43–19.09%, respectively.

**Table 4.** The average recoveries (A.R.), relative standard deviations (RSDs) and limits of quantification (LOQs) of TFP and its metabolites.

| Compound | Matrix | Spiked Level (mg kg$^{-1}$, mg L$^{-1}$) | A.R. (%, *n* = 5) | RSDs (%) | LOQ$_S$ (mg kg$^{-1}$, mg L$^{-1}$) |
|---|---|---|---|---|---|
| TFP | Fresh tea shoots | 10 | 89.02 | 6.72 | 0.005 |
| | | 1 | 90.11 | 8.36 | |
| | | 0.05 | 103.76 | 11.41 | |
| | | 0.005 | 109.90 | 2.47 | |
| | Green tea | 10 | 94.26 | 4.67 | 0.005 |
| | | 1 | 78.03 | 16.00 | |
| | | 0.05 | 98.46 | 13.44 | |
| | | 0.005 | 84.58 | 7.70 | |
| | Black tea | 10 | 101.05 | 4.64 | 0.005 |
| | | 1 | 99.95 | 5.66 | |
| | | 0.05 | 75.38 | 19.09 | |
| | | 0.005 | 82.92 | 12.14 | |
| | Green tea infusion | 0.1 | 96.69 | 7.37 | 0.0005 |
| | | 0.01 | 82.91 | 5.01 | |
| | | 0.005 | 108.77 | 10.04 | |
| | | 0.0005 | 113.99 | 4.58 | |
| | Black tea infusion | 0.1 | 94.58 | 5.89 | 0.0005 |
| | | 0.01 | 114.04 | 12.14 | |
| | | 0.005 | 94.85 | 4.64 | |
| | | 0.0005 | 103.92 | 11.60 | |

| Compound | Matrix | Spiked Level (mg kg$^{-1}$, mg L$^{-1}$) | A.R. (%, $n = 5$) | RSDs (%) | LOQ$_S$ (mg kg$^{-1}$, mg L$^{-1}$) |
|---|---|---|---|---|---|
| PT-OH | Fresh tea shoots | 10 | 90.03 | 6.15 | 0.005 |
| | | 1 | 104.86 | 7.74 | |
| | | 0.05 | 87.18 | 2.04 | |
| | | 0.005 | 94.78 | 7.87 | |
| | Green tea | 10 | 88.72 | 3.33 | 0.005 |
| | | 1 | 78.28 | 1.43 | |
| | | 0.05 | 94.32 | 12.55 | |
| | | 0.005 | 91.61 | 10.09 | |
| | Black tea | 10 | 103.63 | 4.64 | 0.005 |
| | | 1 | 107.69 | 5.22 | |
| | | 0.05 | 87.73 | 6.14 | |
| | | 0.005 | 77.62 | 9.75 | |
| | Green tea infusion | 0.1 | 98.07 | 5.45 | 0.0005 |
| | | 0.01 | 81.28 | 7.86 | |
| | | 0.005 | 98.64 | 9.48 | |
| | | 0.0005 | 101.23 | 6.01 | |
| | Black tea infusion | 0.1 | 97.59 | 8.12 | 0.0005 |
| | | 0.01 | 101.86 | 9.28 | |
| | | 0.005 | 94.27 | 7.29 | |
| | | 0.0005 | 111.96 | 13.83 | |
| PT-CA | Fresh tea shoots | 10 | 91.40 | 10.87 | 0.005 |
| | | 1 | 92.27 | 4.67 | |
| | | 0.05 | 87.07 | 4.89 | |
| | | 0.005 | 96.13 | 7.31 | |
| | Green tea | 10 | 91.52 | 5.12 | 0.005 |
| | | 1 | 80.08 | 5.25 | |
| | | 0.05 | 98.94 | 15.54 | |
| | | 0.005 | 77.46 | 17.84 | |
| | Black tea | 10 | 104.28 | 4.02 | 0.005 |
| | | 1 | 101.20 | 5.35 | |
| | | 0.05 | 83.92 | 6.79 | |
| | | 0.005 | 93.91 | 16.20 | |
| | Green tea infusion | 0.1 | 104.03 | 13.76 | 0.0005 |
| | | 0.01 | 84.74 | 4.21 | |
| | | 0.005 | 102.76 | 9.67 | |
| | | 0.0005 | 114.97 | 8.40 | |
| | Black tea infusion | 0.1 | 98.35 | 7.03 | 0.0005 |
| | | 0.01 | 104.47 | 9.32 | |
| | | 0.005 | 89.44 | 9.84 | |
| | | 0.0005 | 112.37 | 17.08 | |
| OH-T-CA | Fresh tea shoots | 10 | 90.21 | 3.83 | 0.05 |
| | | 1 | 86.15 | 4.40 | |
| | | 0.05 | 96.48 | 13.67 | |
| | Green tea | 10 | 86.64 | 2.76 | 0.05 |
| | | 1 | 76.58 | 3.34 | |
| | | 0.05 | 109.90 | 8.54 | |
| | Black tea | 10 | 105.20 | 6.45 | 0.05 |
| | | 1 | 101.20 | 5.35 | |
| | | 0.05 | 90.36 | 16.24 | |
| | Green tea infusion | 0.1 | 102.09 | 6.96 | 0.005 |
| | | 0.01 | 103.48 | 5.62 | |
| | | 0.005 | 81.78 | 18.50 | |
| | Black tea infusion | 0.1 | 89.91 | 18.03 | 0.005 |
| | | 0.01 | 108.98 | 8.48 | |
| | | 0.005 | 100.21 | 16.98 | |

**Table 4.** *Cont.*

| Compound | Matrix | Spiked Level (mg kg$^{-1}$, mg L$^{-1}$) | A.R. (%, $n = 5$) | RSDs (%) | LOQ$_S$ (mg kg$^{-1}$, mg L$^{-1}$) |
|---|---|---|---|---|---|
| CA-T-CA | Fresh tea shoots | 10 | 84.46 | 2.96 | 0.05 |
| | | 1 | 80.64 | 2.69 | |
| | | 0.05 | 79.60 | 3.34 | |
| | Green tea | 10 | 82.61 | 9.14 | 0.05 |
| | | 1 | 76.33 | 3.51 | |
| | | 0.05 | 86.91 | 15.97 | |
| | Black tea | 10 | 95.31 | 4.10 | 0.05 |
| | | 1 | 90.57 | 5.11 | |
| | | 0.05 | 75.85 | 3.38 | |
| | Green tea infusion | 0.1 | 94.68 | 4.72 | 0.005 |
| | | 0.01 | 85.21 | 11.15 | |
| | | 0.005 | 105.49 | 6.03 | |
| | Black tea infusion | 0.1 | 99.78 | 13.20 | 0.005 |
| | | 0.01 | 104.91 | 4.75 | |
| | | 0.005 | 87.84 | 3.07 | |

As proposed, the LOQs for TFP, PT-OH and PT-CA were 0.005 mg kg$^{-1}$ in fresh tea shoots and dry tea, 0.0005 mg kg$^{-1}$ in tea infusion, while for OH-T-CA and CA-T-CA, the LOQs were 0.05 mg kg$^{-1}$ in fresh tea shoots and dry tea, 0.005 mg kg$^{-1}$ in tea infusion. The method validation results suggested that the developed method was reliable for the simultaneous determination of TFP and its 4 metabolites in tea matrices.

*3.4. Method Application*

Once the analytical method had been established, it was applied to simultaneous determination of TFP and its metabolites in 40 samples (20 green tea and 20 black tea samples), randomly purchased from market. The results were summarized in Table 5, out of the 5 validated compounds, TFP and two of its metabolites, PT-OH and PT-CA were detected during the survey, a large percentage of TFP positive samples were detected, accounting for 70.0% (14 of 20) in green tea and 50.0% (10 of 20) in black tea, respectively. For now, JMPR defined the residue for compliance with the maximum residue limit (MRL) and for dietary risk assessment for plant commodities as TFP [4]. The MRL of TFP in tea is 50 mg kg$^{-1}$ in China, 30 mg kg$^{-1}$ in America, Korea, Japan and 0.01 mg kg$^{-1}$ in EU [38–42]. Besides, as shown in Table 5, the detection of metabolites PT-OH and PT-CA was related to the overdose of TFP, and the higher dosage of TFP, the more residual concentration of metabolites were detected. Considering the toxicity of the metabolites, positive samples with high concentrations of parent compound as well as metabolites detected could pose a greater risk on tea consumers. Thus, in order to control the pesticide residue in tea and minimize the risk, an appropriate usage of the pesticides was urgently needed in tea plantation.

**Table 5.** The residual level of TFP and its metabolites found in different batches of green and black tea samples (mg kg$^{-1}$).

| Sample | No. | Compound (mg kg$^{-1}$) | | | | |
|---|---|---|---|---|---|---|
| | | TFP | PT-OH | PT-CA | OH-T-CA | CA-T-CA |
| Green tea | 1 | 0.118 | n.d. | 0.008 | n.d. | n.d. |
| | 2 | 0.045 | n.d. | n.d. | n.d. | n.d. |
| | 3 | 0.009 | n.d. | n.d. | n.d. | n.d. |
| | 4 | 0.057 | n.d. | 0.011 | n.d. | n.d. |
| | 6 | 0.060 | n.d. | n.d. | n.d. | n.d. |
| | 7 | 0.040 | n.d. | n.d. | n.d. | n.d. |
| | 8 | 0.074 | n.d. | n.d. | n.d. | n.d. |
| | 10 | 0.094 | n.d. | n.d. | n.d. | n.d. |
| | 12 | 0.017 | n.d. | n.d. | n.d. | n.d. |
| | 14 | 0.048 | n.d. | n.d. | n.d. | n.d. |
| | 15 | 0.013 | n.d. | n.d. | n.d. | n.d. |
| | 18 | 0.049 | n.d. | n.d. | n.d. | n.d. |
| | 19 | 0.056 | n.d. | n.d. | n.d. | n.d. |
| | 20 | 0.055 | n.d. | n.d. | n.d. | n.d. |

**Table 5.** *Cont.*

| Sample | No. | Compound (mg kg⁻¹) | | | | |
|---|---|---|---|---|---|---|
| | | TFP | PT-OH | PT-CA | OH-T-CA | CA-T-CA |
| Black tea | 2 | 0.087 | n.d | 0.007 | n.d. | n.d. |
| | 3 | 0.227 | n.d. | 0.040 | n.d. | n.d. |
| | 4 | 0.010 | n.d | n.d | n.d. | n.d. |
| | 5 | 0.029 | n.d | n.d | n.d. | n.d. |
| | 6 | 2.223 | 0.035 | 0.222 | n.d. | n.d. |
| | 7 | 0.075 | n.d. | n.d. | n.d. | n.d. |
| | 8 | 0.028 | n.d. | n.d. | n.d. | n.d. |
| | 10 | 0.025 | n.d. | n.d. | n.d. | n.d. |
| | 19 | 0.251 | n.d. | n.d. | n.d. | n.d. |
| | 20 | 0.013 | n.d | n.d | n.d. | n.d. |

Note: n.d. represents ≤ LOQ.

## 4. Conclusions

In the present study, a series optimization experiments were conducted to establish the method for simultaneous determination of TFP and its metabolites in various tea matrices. As a result, 1%-FA-MeCN was optimized for sample extraction, with a combination of 50 mg $C_{18}$, 50 mg $MgSO_4$, 20 mg GCB and 20 mg $C_{NT}$-OH for clean up. The method was validated to be practicable in analytical performance with excellent linearity, high sensitivity, satisfactory recovery and good precision. The method was successfully applied for the quantitative analysis of TFP as well as its metabolites in market samples. As a result, 24 of 40 tea samples were detected positive for TFP, with PT-CA and PT-OH detected in some of the TFP excessive samples. As far as we know, the analytical method for simultaneous determination of TFP and its metabolites were firstly developed in the present study. The metabolites PT-OH and PT-CA were monitored in tea samples at the first time. The method could be used for routine monitoring as well as comprehensive risk assessment of TFP with its metabolites in tea products.

**Supplementary Materials:** The following supporting information can be downloaded at: https://www.mdpi.com/article/10.3390/agronomy12102324/s1, Figure S1: The intensity of TFP and its metabolites under different cone voltages; Figure S2: The intensity of parent and daughter ions of TFP and its metabolites under different collision energies; Figure S3: The intensity of TFP and its metabolites in different mobile phases (A+B); Figure S4: The recoveries of TFP and its metabolites in different soaking solvents; Figure S5: The recoveries of TFP and its metabolites under different types and quantities of purification adsorbents; Figure S6: The samples after pretreatment (vial 1: green tea sample without purifying; vial 2: green tea sample with purifying by 50 mg $C_{18}$ + 50 mg $MgSO_4$; vial 3: green tea sample with purifying by 50 mg $C_{18}$ + 50 mg $MgSO_4$ + 20 mg GCB + 20 mg $C_{NT}$-OH; vial 4: green tea infusion sample without purifying).

**Author Contributions:** Z.W. and X.W.: method development and manuscript writing; M.W. and Z.L. (Ziqiang Li): method validation.; X.Z., L.Z., H.S., M.Y. and Z.L. (Zhengyun Lou): writing—review & editing; Z.C.: supervision and adviser; F.L.: research design. All authors have read and agreed to the published version of the manuscript.

**Funding:** This study was supported by National Natural Science Foundation of China (No. 32001950), Zhejiang Provincial Natural Science Foundation (LQ19C160016), Innovative Program of the Chinese Academy of Agricultural Sciences (CAAS-ASTIP-TRICAAS).

**Data Availability Statement:** Not applicable.

**Conflicts of Interest:** The authors declare no conflict of interest.

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
