# Peer review of "Establishment of a QuEChERS-UPLC-MS/MS Method for Simultaneously Detecting Tolfenpyrad and Its Metabolites in Tea"

_agronomy, doi:10.3390/agronomy12102324_

Round 1

Reviewer 1 Report

The paper in present form is not sutable for publication. The basis of this statement are following remarks. 

L. 107. What acronyms AHA and AR mean?

L. 112. Acronym in this line is different as in l. 144 and 267

L. 159-163. what are the mobile phases  A and B? Describing  a gradient we start with phase A and than the amount of phase B is changed

L. 222. Why the amount of AA is given in mmol and not in %

L. 225. What are optimal responses?

L. 246. Instead of PHs should be pHs

L. 258. Adsorbability - does exist such a word?

L. 258. What mean the acronyms PWAX, SCX, PSA, C18-N?

L. 284. What are ideal recoveries?

Point 3.3.1. How was LOD determined?

L. 330-331. What mean the descriptions -20%-20% and -50%-50%?

l. 339. The expression "external matrix-matched calibration curves" should be clarified

L. 350. LOQ may be not proposed, it shoud be determined

Point 3.3.2. Between LOD and LOQ exist correlation LOQ = 3LOD. In this work there is no such correlation - why? The units of LOD are mg and units of LOQ are mg/kg and mg/L - why? This important part of the paper is very wrong and it may influence on  the results of it.

L. 375. The acronym MRL should be clarified

L. 384. What is "scientific use"?

Point 4. Only lines 415-421 contain conclusoin. The other part of the point is an additional abstract of the paper.

I was not able to open the supplementary material. In the main text of the paper the fig 1 from the supplement is not mentioned.

Author Response

Major comments:

  1. 107. What acronyms AHA and AR mean?

Response: Sorry for my negligence. The AHA and AR match the abbreviation of ammonium hydroxide solution and analytical reagent, respectively. We’ve added the fully description in 2.1.

  1. 112. Acronym in this line is different as in l. 144 and 267.

Response: Sorry for my error of subscript setting, we have made the correction in the revised manuscript.

  1. 159-163. what are the mobile phases A and B? Describing a gradient we start with phase A and than the amount of phase B is changed.

Response: Thanks for your suggestion. In this study, the 0.1% formic acid in methanol and 0.1% formic acid in water were developed as mobile phases A and B respectively. The gradient started with 10%A and 90%B, we have added the information in the revised manuscript.

  1. 222. Why the amount of AA is given in mmol and not in %?

Response: Sorry for my units error. We prepare mobile phase solution by weighing solid ammonium acetate, so the correction should be mmol L-1. we have revised it in the revised manuscript.

  1. 225. What are optimal responses?

Response: Sorry for my inappropriate expression. Optimal responses just mean the highest intensity in UPLC-MS/MS analysis. we have corrected this statement in the revised manuscript.

  1. 246. Instead of PHs should be pHs

Response: Sorry for my writing error, we have made the corrections.

  1. 258. Adsorbability - does exist such a word?

Response: Thanks for your indication. The correct word should be ''adsorbility'', instead of ''adsorbability''. we have revised it in the revised manuscript.

  1. 258. What mean the acronyms PWAX, SCX, PSA, C18-N?

Response: Tanks for your indication. The PWAX, SCX, PSA, C18-N were abbreviate of some cleaner., corresponding to polymer weak anion exchange, strong cation exchange, primary-secondary amine and octadecyl carbon silica gel.

  1. 284. What are ideal recoveries?

Response: Generally, the recoveries ranged in 75%-120% are regarded as ideal recoveries in pesticide residue studies.

  1. Point 3.3.1. How was LOD determined?

Response: referring to the intensity of the minimum concentration level in linear range, the signal-to-noise (S/N) of 3 was determined as the limit of detection (LODs).

  1. 330-331. What mean the descriptions -20%-20% and -50%-50%?

Response: -20%-20% and -50%-50% are ranges to evaluate the matrix effect, which was calculated by the following formula : ME (%) = (amatrix-asolvent) / asolvent. Where amatrix and asolvent were the slope of the standard curves for matrix-matched calibration and solvent-only calibration, respectively. the descriptions have been revised to make it clear in the revised manuscript.

  1. L. 339. The expression "external matrix-matched calibration curves" should be clarified.

Response: Thanks for your suggestion. Because of the matrix effects of the tea matrices in the present study, a series of matrix-matched calibration standards were used for quantification. The descriptions have been revised to make it clear in the revised manuscript.

  1. 350. LOQ may be not proposed, it should be determined

Point 3.3.2. Between LOD and LOQ exist correlation LOQ = 3LOD. In this work there is no such correlation - why? The units of LOD are mg and units of LOQ are mg/kg and mg/L - why? This important part of the paper is very wrong and it may influence on the results of it.

Response: We are very grateful to your valuable comments. while LOD was generally used to evaluate the response of the analytes on the instrument, so in the present study, the LODs was defined by the signal-to-noise (S/N) of 3 referring to the intensity of the minimum concentration level in linear range of the solvent standard. And in pesticide residue studies, for evaluation of the developed method, matrices should be taken into consideration, so the limit of quantitation (LOQs) were generally defined as the minimum spiked level to meet the requirements of recovery and relative standard deviations (RSDs). Sorry for my mistakes on the units of LOD, we have made the correction in the revised manuscript.

  1. 375. The acronym MRL should be clarified

Response: Thanks for your advice. We have noted the full description at first mention.

  1. 384. What is "scientific use"?

Response: Sorry for my improper expression. We just want to say appropriate use of the pesticide was important, and we have made correction in the revised manuscript.

  1. Point 4. Only lines 415-421 contain conclusio The other part of the point is an additional abstract of the paper.

Response: Thank you for pointing out this problem. we have rewritten this part.

  1. I was not able to open the supplementary material. In the main text of the paper the fig 1 from the supplement is not mentioned.

Response: Thanks for your suggestion. I will upload the supplementary material again, and the the fig 1 from the supplement was mentioned in section 3.1.1, marked Suppl. Figure S1.

Thanks again for all your comments.

Reviewer 2 Report

Journal: Agronomy (ISSN 2073-4395)

Manuscript ID: agronomy-1909803

Type: Article

Title: Establishment of a QuEChERS-UPLC-MS/MS method for simultaneously detecting tolfenpyrad and its metabolites in tea

Overall Recommendations: Accept after Minor Revisions

The quality safety supervision and test of agricultural products urgently need a very excellent analytical method with simultaneously detecting many components in order to assess, prevent and control pesticide and other contaminants residues. The present research work explore the establishment of a modified  QuEChERS  method  with  UPLC-MS/MS  for  simultaneous  determination  of  tolfenpyrad  and  its  metabolites  in  various  tea  matrices 

Abstract

? All the abbreviations must be fully described at first mention along with brackets.

? Problem statement / need for the project / importance of the present research work is found missing.

? Graphical abstract if possible, can make more clear and interesting for the reader.

Keywords

? Keywords should be words not used in the title.

Introduction

? Reference/source missing in Table 1. The chemical structure and toxicological parameters of TFP and its main metabolites. Must be cited with reference or must take copy right permission and submitted to the editor, if applicable.

? Insert references after the first sentences of each paragraph.

? Every paragraph/section needs a who did what statement to end it.

Materials and methods

? Please use standard way of writing units such as mg L-1 instead of mg/L, mg Kg-1 instead of mg/kg.

? All manufacturers have (City, State) after mention.

? Quality assurance of data is mandatory!!! How many batch, repeats, chemical grade and for used instruments manufacturers’ user manual and instructions were strictly followed or not!!!

? Use www.turnitin.com to find and eliminate unnecessary self-repetition and any copied text.

Results and discussion

? Balance out the plurals.

? Split sentences longer than 2 lines into 2-3.

? Do start sentences with The… on occasion.

? Don’t start a sentence with an abbreviation here!

? The text has many typing and grammatical errors, capitalization issues.

 ? English style and language requires a profound revision. However, the readability of the manuscript needs to be improved, preferably carefully reviewing by a native English speaker?

Conclusion

? Novelty of present research work is questionable with reference to practical significance and economic feasibility must be worked and mentioned.

References

? A few very old references have been used. These must be updated with recent research findings or removed. Proper formatting is questionable? It must be according to MDPI Agronomy Journal.

? References formatting are inconsistent. A few DOI are missing? Retrieve date missing? Verify each reference from original source and cross check references in the text and reference section.

Author Response

Minor comments:

  1. Abstract

(1)All the abbreviations must be fully described at first mention along with brackets.

(2)Problem statement / need for the project / importance of the present research work is found missing.

(3)Graphical abstract if possible, can make more clear and interesting for the reader.

Response: Thanks for your value comments very much. We have added the full description of the abbreviations at first mention according to your advice,. Besides, the abstract was rewritten to make the “Problem statement / need for the project / importance of the present research work” more clear.. And Our graphical abstract was as follow:

  1. Keywords
  • Keywords should be words not used in the title.

Response: Thanks for the suggestions. The keywords have been replaced in the revised manuscript.

  1. Introduction
  • Reference/source missing in Table 1. The chemical structure and toxicological parameters of TFP and its main metabolites. Must be cited with reference or must take copy right permission and submitted to the editor, if applicable.

(2) Insert references after the first sentences of each paragraph.

(3) Every paragraph/section needs a who did what statement to end it.

Response: Thanks for your comments. According to your detailed suggestion on introduction, we have noted the reference of the chemical structure and toxicological parameters of TFP and its main metabolites in Table 1. In addition, every paragraph was started and finished insert reference to state.

  1. Materials and methods

(1) Please use standard way of writing units such as mg L-1 instead of mg/L, mg Kg-1 instead of mg/kg.

(2) All manufacturers have (City, State) after mention.

(3) Quality assurance of data is mandatory!!! How many batch, repeats, chemical grade and for used instruments manufacturers’ user manual and instructions were strictly followed or not!!!

(4) Use www.turnitin.com to find and eliminate unnecessary self-repetition and any copied text.

Response: Thank you for pointing out these problems. I have revised all the  units and added the missing information of manufacturers. As for quality assurance, the recoveries of sample added and retrieving experiment (at least three concentrations of high, medium and low, 5 repeats) were conducted. And for used instruments, manufacturers’ user manual and instructions were strictly followed. We have added these statements in the revised manuscript. In addition, according to your advice, we have revised the manuscript thoroughly referring to the duplicate checking report.

  1. Results and discussion

(1) Balance out the plurals.

(2) Split sentences longer than 2 lines into 2-3.

(3) Do start sentences with The… on occasion.

(4) Don’t start a sentence with an abbreviation here!

(5) The text has many typing and grammatical errors, capitalization issues.

(6) English style and language requires a profound revision. However, the readability of the manuscript needs to be improved, preferably carefully reviewing by a native English speaker

Response: We are very grateful to your positive comments detail suggestions for improving our manuscript. We’ve made the corresponding corrections, and according to your advice, the manuscript has been carefully reviewed by a native English speaker in our group and revised thoroughly.

  1. Conclusion

(1) Novelty of present research work is questionable with reference to practical significance and economic feasibility must be worked and mentioned.

Response: Thanks for your advice, we have rewritten the conclusion part and emphasized the novelty and practical significance in the revision.

  1. References
  • A few very old references have been used. These must be updated with recent research findings or removed. Proper formatting is questionable? It must be according to MDPI Agronomy Journal.
  • References formatting are inconsistent. A few DOI are missing? Retrieve date missing? Verify each reference from original source and cross check references in the text and reference section.

Response: Thanks for your detailed suggestion on reference. We have removed some old references and added some recent ones. The format has been readjusted referring to MDPI Agronomy Journal. In addition, we have supplemented the missing DOI and reference date by retrieving.

Thanks again for your comments on this manuscript.

Round 2

Reviewer 1 Report

The new version of the paper is much better than the previous one and it is good enough for publication.